# Genetic Diversity of Enteric Viruses in Children under Five Years Old in Gabon

**DOI:** 10.3390/v13040545

**Published:** 2021-03-24

**Authors:** Gédéon Prince Manouana, Paul Alvyn Nguema-Moure, Mirabeau Mbong Ngwese, C.-Thomas Bock, Peter G. Kremsner, Steffen Borrmann, Daniel Eibach, Benjamin Mordmüller, Thirumalaisamy P. Velavan, Sandra Niendorf, Ayola Akim Adegnika

**Affiliations:** 1Institute of Tropical Medicine, Universitätsklinikum Tübingen, 72074 Tübingen, Germany; manouanacermel@gmail.com (G.P.M.); alvynhyou@gmail.com (P.A.N.-M.); mngwese@gmail.com (M.M.N.); bockc@rki.de (C.-T.B.); peter.kremsner@uni-tuebingen.de (P.G.K.); steffen.borrmann@uni-tuebingen.de (S.B.); velavan@medizin.uni-tuebingen.de (T.P.V.); aadegnika@gmail.com (A.A.A.); 2Centre de Recherches Médicales de Lambaréné (CERMEL), Lambaréné BP 242, Gabon; 3Division of Viral Gastroenteritis and Hepatitis Pathogens and Enteroviruses, Department of Infectious Diseases, Robert Koch Institute, 13353 Berlin, Germany; 4German Center for Infection Research (DZIF), 72074 Tübingen, Germany; 5Infectious Disease Epidemiology, Bernhard Nocht Institute for Tropical Medicine, 20359 Hamburg, Germany; eibach@bnitm.de; 6Department of Medical Microbiology, Radboudumc, 6524 GA Nijmegen, The Netherlands; benjamin.mordmueller@gmail.com; 7Vietnamese-German Center for Medical Research (VG-CARE), Hanoi 100000, Vietnam; 8Faculty of Medicine, Duy Tan University, Da Nang 550000, Vietnam

**Keywords:** enteric viruses, children, phylogenetic analysis, diarrhea, Gabon

## Abstract

Enteric viruses are the leading cause of diarrhea in children globally. Identifying viral agents and understanding their genetic diversity could help to develop effective preventive measures. This study aimed to determine the detection rate and genetic diversity of four enteric viruses in Gabonese children aged below five years. Stool samples from children <5 years with (*n* = 177) and without (*n* = 67) diarrhea were collected from April 2018 to November 2019. Norovirus, astrovirus, sapovirus, and aichivirus A were identified using PCR techniques followed by sequencing and phylogenetic analyses. At least one viral agent was identified in 23.2% and 14.9% of the symptomatic and asymptomatic participants, respectively. Norovirus (14.7%) and astrovirus (7.3%) were the most prevalent in children with diarrhea, whereas in the healthy group norovirus (9%) followed by the first reported aichivirus A in Gabon (6%) were predominant. The predominant norovirus genogroup was GII, consisting mostly of genotype GII.P31-GII.4 Sydney. Phylogenetic analysis of the 3CD region of the aichivirus A genome revealed the presence of two genotypes (A and C) in the study cohort. Astrovirus and sapovirus showed a high diversity, with five different astrovirus genotypes and four sapovirus genotypes, respectively. Our findings give new insights into the circulation and genetic diversity of enteric viruses in Gabonese children.

## 1. Introduction

Gastroenteritis (GE) remains a major public health issue worldwide, especially among children [1]. Globally, more than 700 million cases of acute gastroenteritis in children below 5 years, and 0.8–2 million deaths, per year, have been estimated [2]. GE can have several causes. Previous studies have identified rotavirus, norovirus, astrovirus, and adenovirus as major viral etiologies of diarrheal illness [3,4,5,6]. In particular, rotavirus group A (RVA) (family Reoviridae) remains the most important etiological agent of severe diarrhea in children under five years of age, with an estimated 215,000 deaths recorded in 2013 [7,8]. In recent decades, data on the epidemiology and disease burden of RVA infection have contributed to the implementation of RVA vaccination (the monovalent Rotarix (GlaxoSmithKline Biologicals, Rixensart, Belgium) and the pentavalent human-bovine reassortant RotaTeq (Merck, Kenilworth, NJ, USA)). Although recommended by the World Health Organization (WHO), these two RVA vaccines have demonstrated a low efficacy in resource-poor countries [9].

Noroviruses (NoVs) are associated with estimated deaths of over 200,000 annually, with an important proportion occurring in children from developing countries [10,11]. NoVs comprise the *Norovirus* genus within the Calciviridae family and are small, round structured viruses, non-enveloped with a positive-sense single-stranded RNA genome of around 7.5 kb consisting of three open reading frames (ORFs). ORF1 encodes six non-structural proteins including the viral polymerase. ORF2 and ORF3 encode the major and minor capsid proteins VP1 and VP2, respectively [12,13]. NoV strains are segregated into ten genogroups (G), of which genogroups GI, GII, GIV, GVIII, and GIX contain primarily human viruses linked with GE. Through phylogenetic analyses, at least forty-eight genotypes infecting humans were identified, with the GII.4 genotype responsible for at least 70% of infections worldwide [14,15].

Other viral agents, including astroviruses (AstV, family Astroviridae), sapoviruses (SaV, family Caliviridae), and aichiviruses (AiV, family Picornaviridae), are present overall in much smaller proportions than NoVs and RVA. AstVs, single-stranded RNA viruses, are an important cause of diarrhea in children. They are classified into ten genotypes, of which eight are “classic” genotypes (HAstV1–HAstV8), and two have been described only recently (HAstV-MBL and HAstV-VA/HOM) [16,17]. HAstVs have been identified worldwide and the overall detection rate among children with acute gastroenteritis ranges between 0% and 20% [18]. Human SaVs were detected in people of all ages in both outbreaks and sporadic cases worldwide, with a median detection rate of 6.2% (with a range from 0.2% to 39%) among children <5 years old in low- and middle-income countries [19]. Human SaVs are classified to seventeen different genotypes belonging to four genogroups (GI, GII, GIV, and GV) [20]. AiVs were suggested to play a role in GE, especially in outbreaks caused by contaminated seafood [21]. Human pathogenic AiVs belonging to the genus *Kobuvirus* were classified as AiV-1 and are divided into three different genotypes (A-C) [22], with little information available about their geographical distribution.

In Sub-Saharan Africa and South-East Asia, where over 90% of diarrhea-related deaths occur among children younger than 5 years [23], routine diagnostic or surveillance for viral etiology of diarrheal diseases is lacking or limited [24]. A previous epidemiological study conducted in 2015 in four cities of Gabon reported the predominant proportion of RVA, followed by human adenovirus (HAdV), NoVs, SaV, and AstV among symptomatic children [25]. However, following up these results as part of a national surveillance program prior to rotavirus vaccination in Gabon is essential to establish a solid evidence base.

In this study, we assessed the genetic diversity of enteric viruses in Gabonese children younger than 5 years.

## 2. Materials and Methods

### 2.1. Sample Collection

In this study, stool samples were collected from children under 5 years of age who resided in one of Gabon’s main cities (Lambaréné) and its surroundings (a dominantly rural area without running water and electricity, and reduced access to the health system) between April 2018 and November 2019. Cases were children presenting at the outpatient departments of the two main hospitals with diarrhea or history of diarrhea within the last 24 h. Subsequently, stool samples were randomly obtained from healthy children living in the same neighborhoods and having the same life conditions (source of drinking water, type of toilet, feeding practice, and material of living house) as those of diarrhea cases. All controls were gender- and age-matched with children with diarrhea. The biological material was immediately transported to the CERMEL laboratory and stored in RNAlater at −20 °C until analysis.

### 2.2. Viral RNA Extraction

RNA was extracted from 140 µL of the stool suspension in RNAlater spiked with an internal extraction- and PCR-control (MS-2 phage) using the QIAamp viral RNA Mini Kit (Qiagen, Hilden, Germany). All steps of the RNA isolation were performed following the manufacturer’s instructions and the viral RNA was eluted in a total volume of 60 µL and stored in aliquots at −70 °C.

### 2.3. PCR Detection and Genotyping

The presence of NoV and SaV was first screened using a single-step reverse transcription real time PCR as described previously [26,27]. In short, 10 µL mastermix containing Superscript^TM^ III Platinum OneStep RT-qPCR System (Invitrogen, Karlsruhe, Germany) and 2 µL virus RNA extracted was used for the detection. All NoV positive samples were subsequently genotyped based on amplification and nucleotide sequencing of the RNA-dependent RNA polymerase gene (RdRp, ORF1) and the capsid gene (ORF2, P2 region) as previously described [28]. SaV positive samples by RT-qPCR were characterized by amplifying a polymerase region of 650 bp using a reverse transcription nested PCR. Briefly, RT-PCR with the first PCR round were performed using OneStep RT-PCR kit (Qiagen, Germany) and HotStar Master Mix Kit (Qiagen, Germany) respectively. For the first round, 2 µL of RNA was used in a final reaction volume of 12.5 µL using the SaV 53a+b and SaV 58 primers, and the second round of PCR was performed using the SaV 55a+b and SaV 58 primers. All primers were used at the final concentration of 500 nM (Appendix A).

For the detection of AiV, a nested RT-PCR was performed amplifying 180 bp of 3CD-region as previously described [29]. Primers AI1 and AI2 were used with the OneStep RT-PCR kit (Qiagen, Germany) for the first PCR round and the primers AI3 and AI4 with the HotStar Master Mix Kit (Qiagen, Germany) for the second PCR round. AiV positive samples were further genotyped by using a nested RT-PCR amplifying a 520 bp fragment of the 3CD region. For the first round, 2 µL of RNA was used in a final reaction volume of 12.5 µL by using OneStep RT-PCR kit (Qiagen, Germany) with the AI68 and AI70 primers. For the second round of PCR, the HotStar Master Mix Kit (Qiagen, Germany) was used with the AI69 and AI71 primers. All primers were used at the final concentration of 200 nM (Appendix A).

All samples were screened for AstV infections by a pan-specific AstV semi-nested RT-PCR using the primers AV89a, AV89b, AV89c, AV90a, AV90b, AV90c, and AV91, as previously described [27]. The further genotyping of AstV positive samples was done by reverse transcription nested PCR as previously described [30] using the primers AV91, AV92a, and AV93 (Appendix A).

### 2.4. Nucleotide Sequencing and Phylogenetic Analysis

All PCR products (NoV, SaV, AiV, and HAstV) were submitted to Sanger sequencing using the corresponding PCR primers (Appendix A). Phylogenetic analysis was performed with Geneious prime 11.0.4 and MEGA7.0.26. Norovirus sequences were submitted to the online Norovirus Tool to assign genotypes (https://www.rivm.nl/mpf/typingtool/norovirus/) (accessed on 1 December 2020). For phylogenetic analysis, nucleotide sequences were aligned with the MAFFT algorithm in Geneious prime 11.0.4. In MEGA7.0.26, trees were constructed using the best fit models of substitution pattern with the lowest BIC score (Bayesian information criterion). The reliability of the branching pattern was tested with bootstrapping (1000 replicates). The nucleotide sequences of the viral pathogens of this study are available under the following GenBank accession numbers AiV MW525344-MW525346, HAstV MW525347-MW525357, NoV-GI MW506839-MW506842 (ORF1), and MW513431-MW513437 (P2 region); NoV-GII MW513401-MW513421 (ORF1) and MW506843-MW506861 (P2 region); and SaV MW525358-MW525362.

### 2.5. Statistical Analysis

Statistical analysis was done by Pearson’s chi-square and Fisher’s exact tests using GraphPad Prism software version 6.00 for Windows. A *p*-value < 0.05 was considered to be statistically significant.

## 3. Results

### 3.1. Study Population

A total of 244 stool specimens were collected from participants aged between 0 and 59 months with a median age of 14 months. Of these, 72.5% (177/244) were symptomatic children between 0 and 59 months of age with a median age of 12 months, whereas the median age of asymptomatic participants was 24 months. Among children with diarrhea, 57.1% (101/177) were male and 70.1% (124/177) lived in an urban area; compared to the healthy group in which 41.8% (28/67) were male and 61.2% (41/67) lived in an urban area (Lambaréné). Overall, 190 children of the study population were aged between 0 and 24 months, with a high proportion of symptomatic participants (81.6%; 155/190).

### 3.2. Detection Rate of Enteric Viruses in the Study Population

Among 177 stool samples collected from patients with diarrhea symptoms, 41 (23.2%) were positive for at least one virus, versus 10/67 (14.9%) in the healthy group (*p* = 0.2). NoV (14.7%; 26/177) was the most prevalent in symptomatic children, followed by AstV (7.3%; 13/177). The two remaining viruses (SaV and AiV) were detected at a lower rate of 3.4% (6/177) and 1.1% (2/177), respectively. The overall detection rate of NoV, AstV, SaV, and AiV was 9% (6/67), 4.5% (3/67), 1.5% (1/67), and 6% (4/67), respectively, among controls. Among the NoVs, NoV-GII was the most frequently detected in patients with diarrhea, whereas the detection rate was similar for NoV-GI and NoV-GII in the control group. There was no statistical significance between detection rate of enteric viruses in symptomatic and healthy children (Table 1). Of the 51 study participants who tested positive for enteric viruses, 17.6% contained more than one virus. Mixed infections were found in 6 (3.4%) children suffering from diarrhea (NoV-GI+SaV, NoV-GI+AstV, NoV-GII+AstV (*n* = 3) and SaV-AstV) and 3 (4.5%) cases in control group (NoV-GI+AiV, NoV GI+SaV and NoV-GII+AstV).

Table 2 summarizes the distribution of enteric viruses in children with/without diarrhea according to age, sex, and living area. Overall, enteric viruses screened were found only in symptomatic children aged 0 to 24 months, whereas the age group 25 to 59 months was found to be more infected among controls. NoV, particularly NoV-GII, was the most frequently detected virus within the six first months of life, with 21.3% (10/47) of children with diarrhea. The age group 7 to 12 months was found to be infected by all four enteric viruses (NoV, AstV, SaV, and AiV) with detection rates of 17% (8/47), 12.8% (6/47), 2.1% (1/47), and 4.3% (2/47), respectively. No case of any enteric viruses tested in this study was found in children from 24 to 59 months presenting with diarrhea. In contrast to the control group, symptomatic children from surrounding villages and female diarrhea cases were more infected with gastroenteritis viruses detected in this study.

The temporal pattern of enteric viruses detected from symptomatic children during the study period is shown in Figure 1. NoV, SaV, and AstV were more frequently detected during the dry season than during the rainy season. There was, however, a significant difference in NoV occurrence between dry and rainy seasons throughout the sampling period (dry season (20.4%; 20/98) and rainy season (7.6%; 6/79) *p* = 0.019), with a notable peak of NoV infection in July 2018 and August 2019. The two cases of AiV were detected during the rainy season. The seasonal distribution of enteric viruses from asymptomatic children is not shown because samples were collected very late in the study period and did not cover at least one year of sampling.

### 3.3. Sequence and Phylogenetic Analyses of Gastroenteritis Viruses

#### 3.3.1. Noroviruses

From 32 norovirus positive samples, 31 (96.9%) were genotyped. Of note, norovirus GII was most frequently detected 71% (22/31) followed by GI at 29% (9/31) (Table 3). Based on both RdRp and capsid sequences, four genotypes were identified among NoV-GII including a rare recombinant GII.P31-GII.4 New Orleans strain; for one sample, only the capsid sequence could be determined (GII.6). Overall, the common recombinant GII.P31-GII.4 Sydney strain was predominant at 45.2% (14/31). Regarding NoV-GI, four strains were genotyped on both RNA polymerase and capsid sequences, among which we found a rare recombinant (GI.P11-GI.2). In addition, four genotypes were only identified based on the capsid gene.

Phylogenetic analysis based on RNA polymerase and capsid regions for GI showed a diversity and clustering among Gabonese strains. The phylogenetic trees obtained in this study demonstrated that all NoV-GI strains genotyped either from RdRp or capsid region were grouped in different clades. Interestingly, all of these NoV-GI Gabonese strains shared a close similarity with recombinants from different geographical areas, indicating the circulation of closely related strains globally (Figure 2).

The phylogenetic analysis for NoV-GII revealed that NoV-GII Gabonese strains formed three distinct phylogenetic clusters in the two trees based on RdRp and capsid sequences, respectively. The polymerase-based tree clearly showed that 17 Gabonese strains were closely related to the reference strain GII.P31 (JX459907). The other clusters were most similar to GII.P17 (LC037415). A similarly clustering was observed with the phylogenetic tree on the capsid region. Of the four sequences which clustered in the ORF1 with the reference strain GII.P17 (LC037415), three samples could be amplified in the P2 region. These three sequences clustered in the P2-phylogenetic tree with the reference sequence GII.17 (AY502009). Of the 17 GII.P31 sequences, 16 samples could be amplified in the P2 region. Three of these sequences clustered with the reference sequence GII.4 New Orleans (JN400623), whereas the other sequences were more related to the reference sequence GII.4 Sydney (JX459908) (Figure 3).

#### 3.3.2. Astroviruses

This study revealed that 6.6% (16/244) of the participants had AstV infection. Sequence and phylogenetic analysis of the positive cases from a partial nucleotide sequence of ORF1b region confirmed 11 HAstV genotypes. This analysis demonstrated that samples belonged to five different HAstV genotypes. Classic HAstV were predominant with a proportion of 63.7% (7/11). Among these, three samples belonged to HAstV-5, three others belonged to HAstV-4, and the last sample belonged to HAstV-8. Within the newly HAstV, three samples belonged to HAstV-VA2 and the remaining sample belonged to MLB type 1 (MLB1) (Figure 4).

#### 3.3.3. Sapoviruses

In order to characterize SaV strains, the sequence of polymerase (partial nucleotide sequence of ORF1) was phylogenetically analyzed (Figure 5). Five of all SaV positive samples were successfully genotyped and assigned into two distinct genogroups, GI and GII. Three different GI genotypes were found in our study: GI.2 (608), GI.3 (317) and GI.1 (C021 and 449). The GII strain was classified into GII.4 (524).

#### 3.3.4. Aichiviruses

In this study, six AiV positive samples were found, confirmed by sequencing. To characterize these samples in more detail we used a prolonged PCR in the same region (3CD). Our analysis revealed that two samples belonged to genotype A and one sample belonged to genotype C, which is rare [21] (Figure 6).

## 4. Discussion

The present study describes the detection and characterization of viral agents implicated in gastroenteritis in Gabonese children. Over the duration of the study, at least one enteric virus was detected in 20.9% of the study population, of which 23.2% (41/177) were symptomatic cases and 14.9% (10/67) in the control group. Contrary to the previous study conducted in Gabon in 2015 [25], rotaviruses and adenoviruses were not tested in this study. In accordance with previous studies, NoV was predominant, followed by HAstV, SaV, and AiV [25,31]. AiV, reported for the first time in Gabon and in Central Africa, was found with a lower detection rate in symptomatic cases (1.1%) than in healthy children (6%). This finding is in line with previous reports, suggesting that AiV may not an important viral gastroenteritis agent [32]. In our study, more than one virus was detected in 17.6% of studied children, while higher detection rates of 35.5% were found in Cameroon, in Gabon (33.7%), and mixed infections rates of 6.7% and 2.8% were respectively reported in Nigeria and Canada [25,33,34,35]. In this regard, we clearly noticed a disparity of mixed infections rate amongst studies, which could be explained by the number of targeted virus species, sample size, sampling (repeat sampling), storage, and the detection methods [34].

Our findings demonstrated that children were carried enteric viruses implicated in diarrhea in the 0 to 24 months age group. This is consistent with previous reports emphasizing that maternal antibodies transmitted during breastfeeding are not sufficient to protect against gastroenteritis viruses during the first 24 months of life [36,37]. Furthermore, our results suggest that the presence of viruses in this age group could be due to direct contact with people who experienced gastroenteritis symptoms, contaminated water, or food [38]. This could be explained by the fact that at this age, children crawl, are carried by people, and are likely to touch any contaminated object.

A significant seasonal pattern of NoV was observed in this study with major peaks during dry seasons. This finding is similar to that reported in Cameroon showing a peak of NoV in the beginning of wet season [33]. HAstV was detected with a major peak during the dry season, which is similar to a report from Nigeria [34]. Regarding SaV and AiV, we observed a relative seasonal distribution throughout the sampling period with a peak during the dry season for SaV and the occurrence of AiV in April and November. These findings are in line with the results described in previous studies reporting SaV seasonal distribution with a peak in the dry season, and high seasonal distribution of AiV infections in January, February, and December [22,39].

In this study, NoV was the most common virus detected in symptomatic children (14.7%) and the control group (9%), with a high rate of NoV-GII among participants with diarrhea (11.3%). This predominance was also reported in other studies showing that NoV was the second most prevalent virus after rotavirus compared to other enteric viruses. In addition, our results are comparable to findings in Cameroonian and Burkinabe studies reporting NoV cases in healthy children, albeit with a considerably lower detection rate in our study [33,40]. More generally, we found a lower prevalence of NoV in our study compared to that reported in all previous studies from neighboring countries [33,37]. As observed elsewhere, GI strains were less frequently detected and had a higher diversity than GII genogroups [26,30,41]. The predominance of the GII.4 variants in this study may reinforce the hypothesis and emphasize the need to explore whether new GII.4 variants possibly arise in Africa where some of the GII.4 variants circulated earlier [24]. Moreover, the fact that NoV GII.4 was found to be predominant in our study, whereas in various African countries the GII.17 strain emerged and was mostly detected in Kenya, underlines that dynamic evolution of NoVs might occur unexpectedly [42,43]. The most prevalent NoV genotype was GII.P31-GII.4 Sydney, with a 45.2% detection rate. This finding is consistent with the results reported in Germany, Brazil, and China [26,44,45]. Surprisingly, we did not find that the emerging recombinant strain GII.P16-GII.4 replaced GII.P31-GII.4 Sydney around the world [46,47].

Our results confirmed that HAstV (7.3%) is less common in children with acute gastroenteritis compared to NoV. This finding is relatively similar to the detection rate of HAstV found in children with diarrhea in Nigeria (6.8%) [30] and Burkina Faso (4.9%) [40]. HAstV showed a high diversity in this study, clustering with newly identified HAstV (HAstV-VA2 and MLB1) and classic HAstV (HAstV5, HAstV4, and HAstV8). Both HAstV genotypes (HAstV-VA2 and MLB1) were for the first time reported in Gabon. Interestingly, we did not find the most common HAstV-1 in this study but only uncommon HAstV-5 and rare HAstV-8 genotypes were identified.

SaV was found at a low detection rate of 3.4% in children with diarrhea. This is in line with a previous report revealing a range of 3% to 17% of SaV infections among children with gastroenteritis in high- and low-income countries [48].This result also shows that SaV is less detected in gastroenteritis cases than NoV. The circulation of SaV genogroups shows variability with SaV-GI. Moreover, SaV-GI.1 was detected in two individuals whereas only one case of each other GI genotypes was detected among the study population. In contrast, SaV-GII was less frequent in this present study than in Burkinabe children with diarrhea where SaV-GII appeared to be predominant [40].

A low prevalence of AiV A strains was found in this study with more cases in healthy children as reported elsewhere [32]. This finding is concordant with several studies that reported low prevalence of this enteric virus [49,50]. All detected AiV strains belonged to genotype A and C. In contrast to previous findings reporting the presence of AiV A in children with gastroenteritis [49,51], the AiVs A detected in our study were from healthy children. This reinforces the need of further studies to elucidate the full spectrum of clinical symptoms of AiV. The rare AiV C reported only in Burkina Faso [40] and France [52] from children with gastroenteritis was also identified in a symptomatic child in our study.

Overall, due to the close relationship between the Gabonese enteric viruses and the reference strains, our results revealed that the same strains are circulating around the world. This could be explained by the migration of foreigners that facilitate the importation of infectious agents [53]. In addition, the relative proportions of viruses reported here suggests a possible problem of life conditions, such as water or food contamination, or the sociodemographic level of population. It is therefore important to understand the local epidemiology of these viruses, which will provide information on the disease transmission and contribute to the development of vaccines and treatment strategies.

A limitation of our study is the small size of control group. Furthermore, the fact that the rotavirus and adenovirus were not screened in this study underestimated the information about virus co-infection. In addition, seasonal distribution of enteric viruses among asymptomatic children was not shown because of the sampling that started towards the second year.

## 5. Conclusions

In conclusion, this study shows an important rate of viral detection (23.2%) among Gabonese children <5 years with diarrhea. NoV was the predominant virus associated with diarrhea. To the best of our knowledge, this study provides the first report on the detection of AiV in Central Africa, particularly in Gabon.

Molecular characterization reported a great diversity of enteric virus strains. Thus, our data would be useful for better management of preventive strategies such as vaccination.

## Figures and Tables

**Figure 1 viruses-13-00545-f001:**
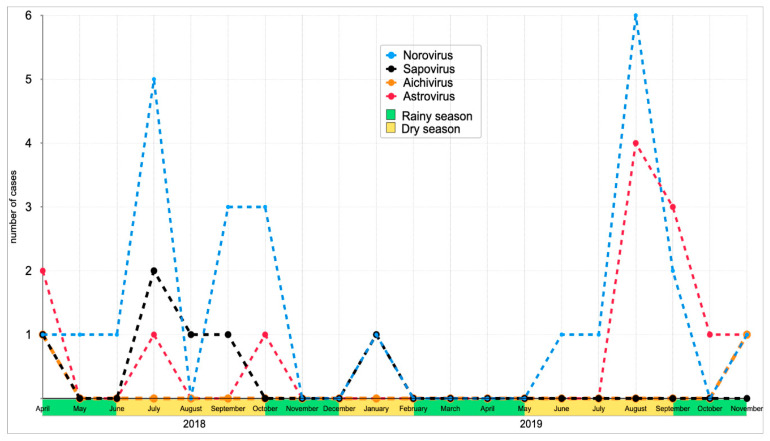
Seasonal distribution of enteric viruses among diarrhea cases within the sampling period, 2018–2019.

**Figure 2 viruses-13-00545-f002:**
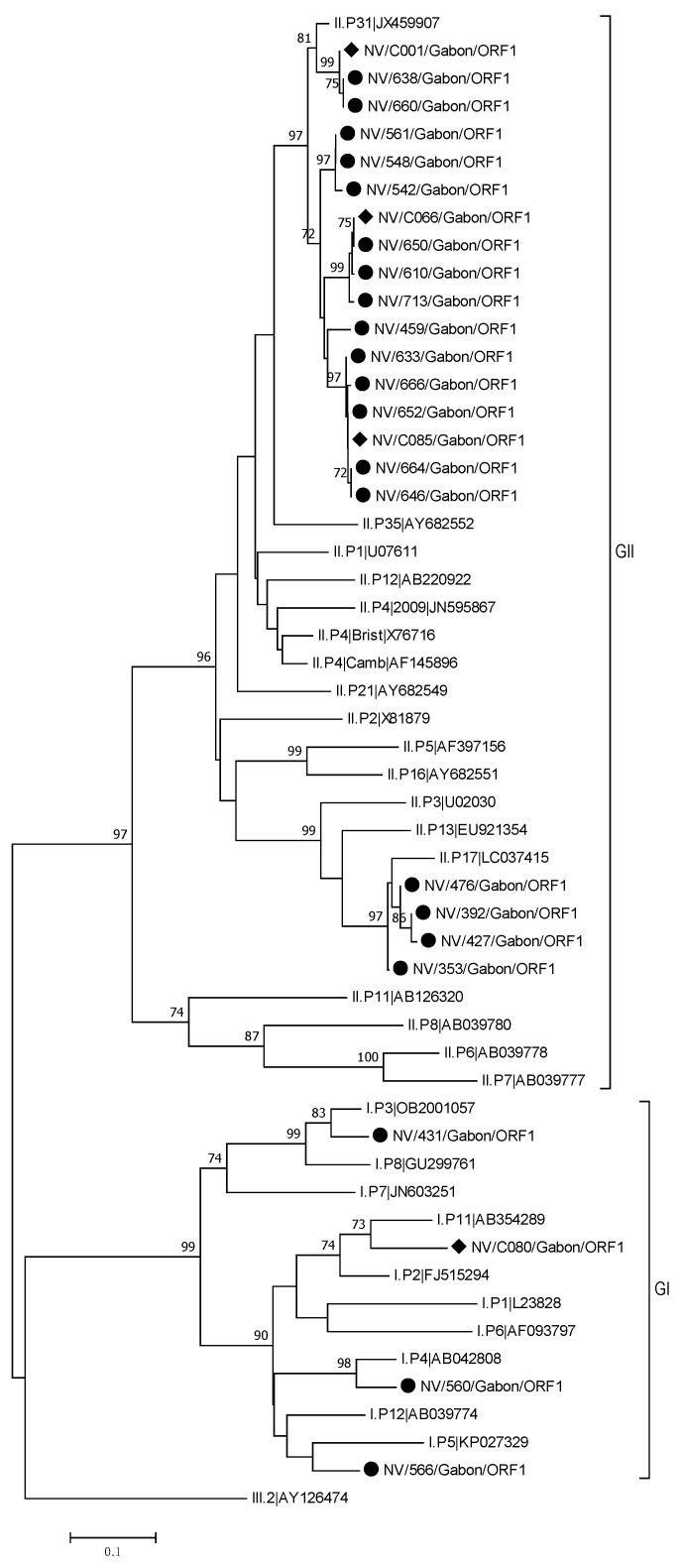
Phylogenetic tree of a 248 bp alignment of ORF1 region of the GI and GII NoV strains from Gabon and NoV reference sequences (accession nos. are indicated). Samples from symptomatic patients are marked with a dot and samples from asymptomatic with a rhombus. The tree was constructed using the neighbor-joining method with bootstrap test (1000 replicates) and the Kimura 2-parameter method available in MEGA7. Bootstrap values above 70 are shown. The bar indicates the nucleotide substitution per site.

**Figure 3 viruses-13-00545-f003:**
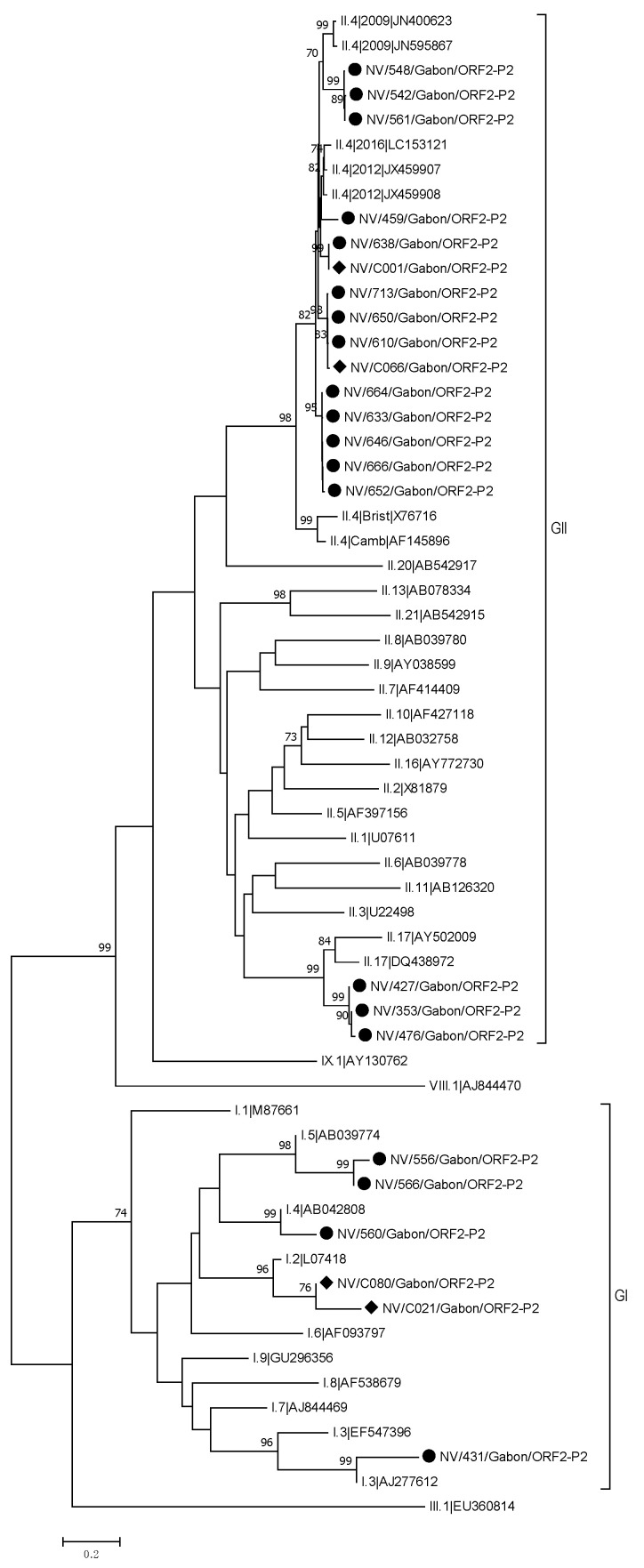
Phylogenetic tree of a 691 bp alignment of P2 region (ORF2) of the GI and GII NoV strains from Gabon and NoV reference sequences (accession nos. are indicated). Samples from symptomatic patients are marked with a dot and samples from asymptomatic patients with a rhombus. The tree was constructed using the neighbor-joining method with the bootstrap test (1000 replicates) and the Kimura 2-parameter method available in MEGA7. Bootstrap values above 70 are shown. The bar indicates the nucleotide substitution per site.

**Figure 4 viruses-13-00545-f004:**
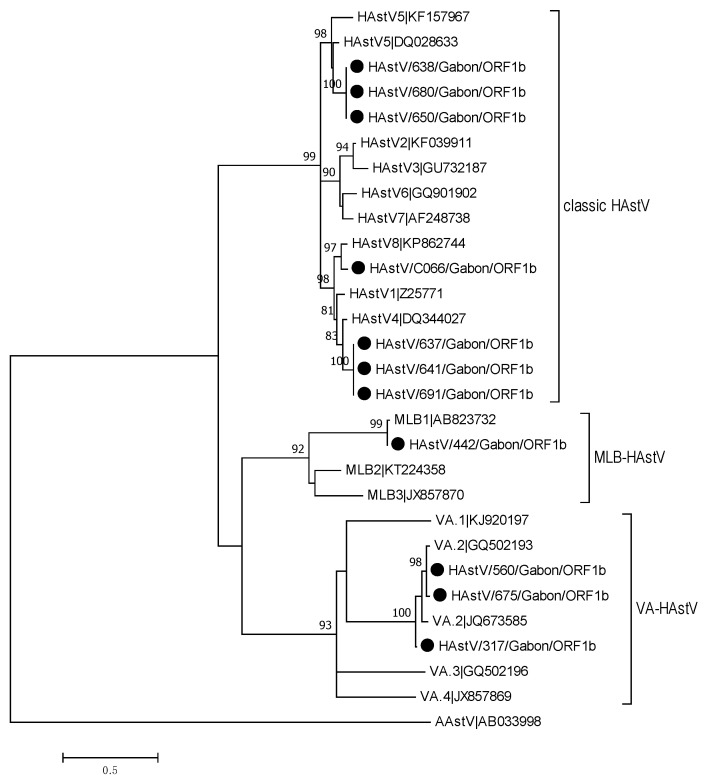
Phylogenetic tree of a 419 bp alignment of ORF1b region of astrovirus strains from Gabon and HAstV reference strains (accession nos. are indicated). Samples from symptomatic patients are marked with a dot and samples from asymptomatic patients with a rhombus. The tree was constructed using the maximum likelihood method with the bootstrap test (1000 replicates) and Tamura 3-parameter method available in MEGA7. The bar indicates the nucleotide substitution per site.

**Figure 5 viruses-13-00545-f005:**
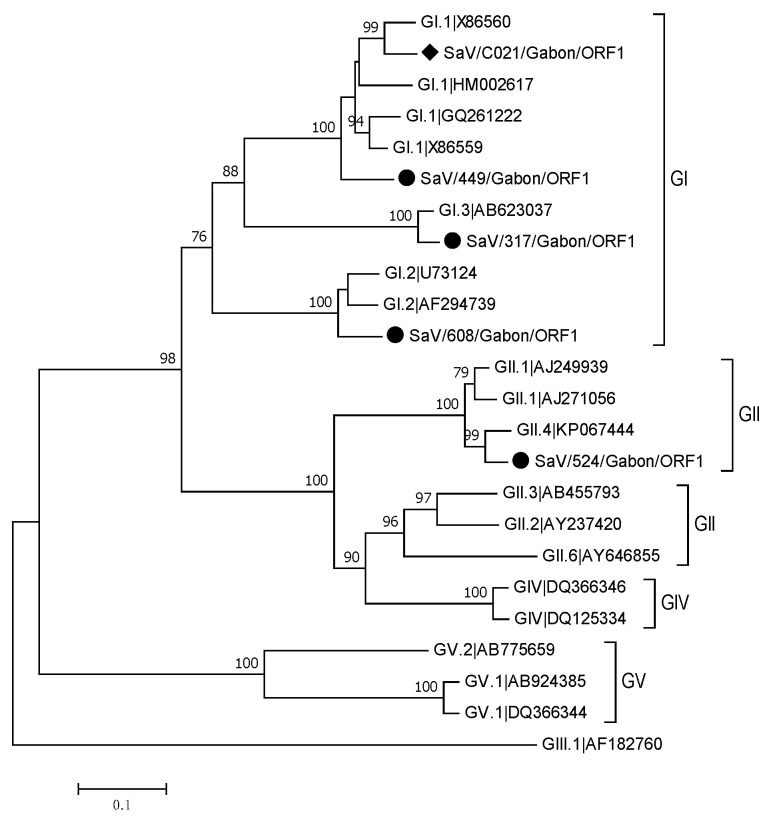
Phylogenetic tree of a 593 bp alignment of polymerase region (ORF1) of sapoviruses (SaV) strains from Gabon and SaV reference sequences (accession nos. are indicated). Samples from symptomatic patients are marked with a dot and samples from asymptomatic patients with a rhombus. The tree was constructed using the neighbor-joining method with the Bootstrap test (1000 replicates) and the Kimura 2-parameter method available in MEGA7. The bar indicates the nucleotide substitution per site.

**Figure 6 viruses-13-00545-f006:**
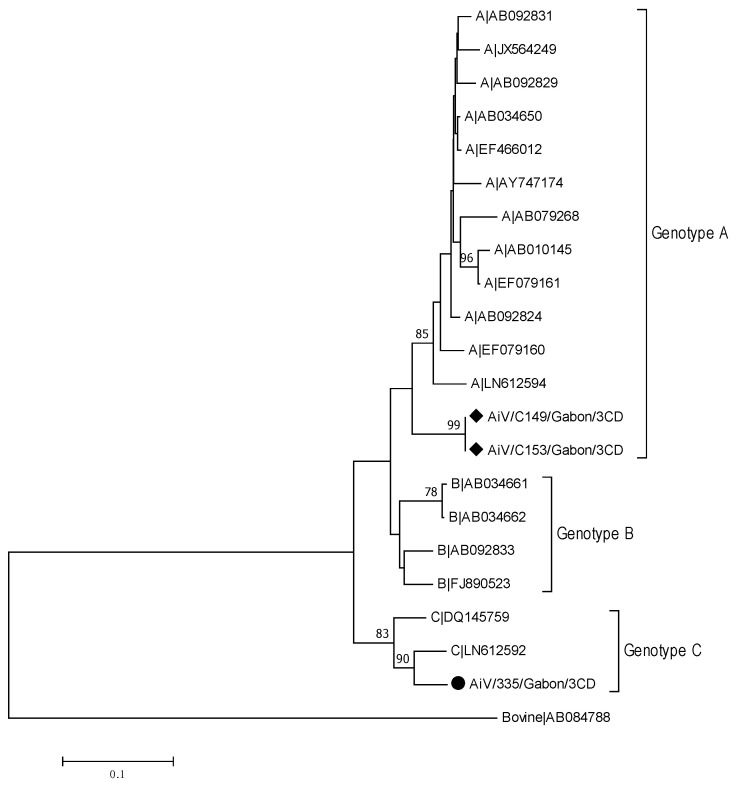
Phylogenetic tree of a 459 bp alignment of 3CD region of aichivirus (AiV) strains from Gabon and AiV reference sequences (accession nos. are indicated). Samples from symptomatic patients are marked with a dot and samples from asymptomatic patients with a rhombus. The tree was constructed using the neighbor-joining method with the bootstrap test (1000 replicates) and Tamura 3-parameter method available in MEGA7. Bootstraps value above 70 are shown. The bar indicates the variation scale.

**Table 1 viruses-13-00545-t001:** Detection rate of enteric viruses among children with/without diarrhea.

Virus	Diarrhea (*n* = 177)	No Diarrhea (*n* = 67)	*p*-Value	Total (*n* = 244)
NoV	26 (14.7%)	6 (9.0%)	0.2764	32 (13.1%)
NoV GI	6 (3.4%)	3 (4.5%)	1.0000	9 (3.7%)
NoV GII	20 (11.3%)	3 (4.5%)	0.1046	23 (9.4%)
AstV	13(7.3%)	3 (4.5%)	0.5371	16 (6.6%)
SaV	6 (3.4%)	1 (1.5%)	0.6212	7 (2.9%)
AiV	2 (1.1%)	4 (6.0%)	0.1184	6 (2.5%)

**Table 2 viruses-13-00545-t002:** Distribution of enteric viruses in the study population according age, gender, and location.

**Diarrhea Cases**						
	**NoV *n* (%)**	**NoV GI *n* (%)**	**NoV GII *n* (%)**	**AstV *n* (%)**	**SaV *n* (%)**	**AiV *n* (%)**
Age group (Months)						
0–6 (*n* = 47)	11 (23.4)	1 (2.1)	10 (21.3)	4 (8.5)	0 (0)	0 (0)
7–12 (*n* = 47)	8 (17.0)	3 (6.4)	5 (10.6)	6 (12.8%)	1 (2.1)	2 (4.3)
13–18 (*n* = 45)	5 (11.1)	2 (4.4)	3 (6.7)	2 (4.4)	2 (4.4)	0 (0)
19–24 (*n* = 16)	2 (12.5)	0 (0)	2 (12.5)	1 (6.3)	3 (18.8)	0 (0)
25–59 (*n* = 22)	0 (0)	0 (0)	0 (0)	0 (0)	0 (0)	0 (0)
Gender						
F (*n* = 76)	13 (17.1)	2 (2.6)	11 (14.5)	9 (11.8)	4 (5.3)	1 (1.3)
M (*n* = 101)	13 (12.9)	4 (4.0)	9 (8.9)	4 (4.0)	2 (2.0)	1 (1.0)
Residence						
Surrounding villages (*n* = 53)	10 (18.9)	3 (5.7)	7 (13.2)	3 (5.7)	4 (7.5)	1 (1.9)
Lambaréné (*n* = 124)	16 (12.9)	3 (2.4)	13 (10.5)	10 (8.1)	2 (1.6)	1 (0.8)
Healthy children						
	**NoV *n* (%)**	**NoV GI *n* (%)**	**NoV GII *n* (%)**	**AstV *n* (%)**	**SaV *n* (%)**	**AiV *n* (%)**
Age group (Months)						
0–6 (*n* = 7)	1 (14.3)	0 (0)	1 (14.3)	0 (0)	0 (0)	0 (0)
7–12 (*n* = 8)	0 (0)	0 (0)	0 (0)	1 (12.5)	0 (0)	0 (0)
13–18 (*n* = 4)	0 (0)	0 (0)	0 (0)	0 (0)	0 (0)	0 (0)
19–24 (*n* = 16)	0 (0)	0 (0)	0 (0)	0 (0)	0 (0)	1 (6.3)
25–59 (*n* = 32)	5 (15.6)	3 (9.4)	2 (6.3)	2 (6.3)	1 (3.1)	3 (9.4)
Gender						
F (*n* = 26)	1 (3.8)	1 (3.8)	0 (0)	1 (3.8)	0 (0)	2 (7.7)
M (*n* = 41)	5 (12.2)	2 (4.9)	3 (7.3)	2 (4.9)	1 (2.4)	2 (4.9)
Residence						
Surrounding villages (*n* = 39)	3 (7.7)	0 (0)	3 (7.7)	2 (5.1)	0 (0)	3 (7.7)
Lambaréné (*n* = 28)	3 (10.7)	3 (10.7)	0 (0)	1 (3.6)	1 (3.6)	1 (3.6)

**Table 3 viruses-13-00545-t003:** Norovirus (NoV) polymerase and capsid genotypes obtained from the study population.

Polymerase (RdRp) Genotype	Capsid (P2 Domain) Genotype	Number (%)
Genogroup I		
GI.P3	GI.3	1 (3.2)
	GI.5	1 (3.2)
GI.P4	GI.4	1 (3.2)
GI.P5	GI.5	1 (3.2)
	GI.3	2 (6.5)
	GI.	1 (3.2)
	GI.2	1 (3.2)
GI.P11	GI.2	1 (3.2)
Genogroup II		
GII.P17	GII.17	3 (9.7)
GII.P17	GII.	1 (3.2)
GII.P31	GII.4 Syd	14 (45.2)
	GII.6	1 (3.2)
GII.P31	GII.4 NO	3 (9.7)

## Data Availability

The data presented in this study are available on request from the corresponding author.

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
