# Peer review of "Genetic Diversity of Enteric Viruses in Children under Five Years Old in Gabon"

_viruses, 2021, doi:10.3390/v13040545_

Round 1

Reviewer 1 Report

This manuscript by Gédéon Prince Manouana et al., determines the atack rate and genetic diversity of norovirus, astrovirus, sapovirus and Aichi virus  A in children <5 years old in Gabon by PCR, sequencing and phylogenetic analyzes,  and concludes that there is an important viral detection rate in these children with diarrhoea, being NoV the most predominant virus associated with. They also report a great diversity between the different virus strains.  This aspect is essential to fill the knowledge gap on these pathogens since prompt and meticulous preventive strategies of vaccination are needed. 

On the other hand, this manuscript has no taken in account rotavirus infection, the main cause of diarrhoea in children under 5 years old, the effects after the rotavirus vaccine implementation, possible co-infections with other viruses. But despite the differences between similar studies and some limitations of this study, this paper is quite interesting and worth of consideration provided the authors can implement the aspects as indicated below.

Main remarks:

Introduction should be improved. Authors should also talk about rotavirus, rotavirus vaccine.

Materials and methods. I would recommend to separate "PCR and sequence analysis" section in different sections, such as "RNA extraction", "PCR detection and genotyping" and "Sequence analyses". I would also suggest to show PCR primers and conditions on tables, maybe as supplementary tables.

Table 2.- Distribution of enteric viruses in the study population according age, gender, and location. I would suggest to the authors to present these data on figures using bar graphics. For exemple, each virus on "y axis" and the age group on "x axis", and in the same bar diarrhoea cases and healthy children. In this way it would be much easier to see the differences between the virus distribution in the study population. 

Phylogenetic analyzes: Authors should improve footnote (meaning of the bars, phylogenetic tree based on..., reference strains used, GenBank accession numbers are listed...) and specify genogroup/genotype on trees. 

Additional experiments. Authors should also analyze at least rotavirus and possible patterns of coinfections of enteric viruses in the stool samples collected. Why rotavirus was not tested??

Minor remarks:

In all figures an tables --> change : into . (Figure 1: --> Figure 1.)

Line 7 to 15: Alignment of the different places and superindex numbers.

Line 42: "it has been estimated that more..." -->"it has been estimated more..."

Line 44: GEcan --> GE can

Line 52: [9], [10] -->  [9,10]

Line 55: "infectious to humans" --> better "infecting humans"

Line 95: Feeding --> feeding

Line 98: -20 ºC --> -20ºC

Line 102: "of the suspension of stool...", better "of the stool suspension..."

Line 105-106: 60 ul together, not in different lines. -70 ºC --> -70ºC

Line 107: "The presence of NoVs..." --> "The presence of NoV..."

Line 110: "and a 2 ul virus RNA extract.." --> "and 2 ul viral RNA extracted.."

Line 114: "of 650bp..." --> of 650 bp..."

Line 138: "infections by using..." --> "infections by..."

Line 139: "RT-PCR by using..." --> "RT-PCR using..."

Line 141: "was done _ by..." -->  "was done by..." 

Line 166: "from participants aged between 0-59 months..."

Line 171: "...lived an urban..." --> "...lived in an urban..."

Line 282: "...sequences of P2-region..." --> "...sequences of P2 region..."

Line 311: "...a 593bp..." --> "...a 593 bp..." 

Line 322: "...was constructed on the base of 459 bp nucleotide sequence of ..."

Line 329: "...of which 23.2% (41/177) of symptomatic..." --> "...of which 23.2% (41/177) were symptomatic..."

Line 390: "All detected the AiV..." --> "All detected AiV..." 

Author Response

Dear Reviewer,

We thank you very much for your appreciation of our study.

Introduction

should be improved. Authors should also talk about rotavirus, rotavirus vaccine.
Reply: Regarding the above comment, we added a small paragraph talking a bit about rotavirus infection (Line 45-53)

Materials and methods

I would recommend to separate "PCR and sequence analysis" section in different sections, such as "RNA extraction", "PCR detection and genotyping" and "Sequence analyses". I would also suggest to show PCR primers and conditions on tables, maybe as supplementary tables.

Reply: We thank the reviewer for this suggestion. All primers were already listed in supplementary Table 1 provided. Therefore, we separated the “Materials and methods” section in different sections as suggested above and we also included changes in the text (lines 127-131 and 137-146). In addition, we provided a supplementary Table 2 in which details of unpublished PCR programs and primers are described.

Table 2.- Distribution of enteric viruses in the study population according age, gender, and location. I would suggest to the authors to present these data on figures using bar graphics. For exemple, each virus on "y axis" and the age group on "x axis", and in the same bar diarrhoea cases and healthy children. In this way it would be much easier to see the differences between the virus distribution in the study population. 

Reply: Presenting our data in a figure as suggested implies loss of important information such as the number of participants by categorical variables (denominator “N”) and the number of those who were tested positive for any virus (numerator “n”). Information that provides a picture of the demographic characteristics of the study population. Thus, the significance of our findings may be lost to non-expert readers. We therefore prefer to leave this table as it stands. Given that Table 2 provides more details which support the study population section, we believe that the use of this is justified.

Phylogenetic analyzes: Authors should improve footnote (meaning of the bars, phylogenetic tree based on..., reference strains used, GenBank accession numbers are listed...) and specify genogroup/genotype on trees.

Reply: Thanks for this very helpful comment, we changed all figure legends accordingly. Genotypes or genogroups were specified on the phylogenetic trees. GenBank accession No. of the reference strains are indicated in the name of the sequence, GenBank accession No. of the analyzed samples are listed in the Material and Methods section. Aichivirus phylogenetic tree was recalculated for clearer separation of the genotypes.

Additional experiments. Authors should also analyze at least rotavirus and possible patterns of coinfections of enteric viruses in the stool samples collected. Why rotavirus was not tested??

Reply: Indeed, data on rotavirus would have an additional impact on our results, which is why the fact that this virus has not been tested has been mentioned in the limitation section. Following your question, we would say we unfortunately did not test it in this study because previous work in progress was devoted to rotavirus in the same study population. 

Minor remarks.

In all figures an tables --> change: into . (Figure 1: --> Figure 1.)

Reply: We changed accordingly

Line 7 to 15: Alignment of the different places and superindex numbers.

Reply: We changed accordingly

Line 42: "it has been estimated that more..." -->"it has been estimated more..."

Reply: We edited accordingly

Line 44: GEcan --> GE can

Reply: We corrected accordingly

Line 52: [9], [10] --> [9,10]

Reply: We edited accordingly (line 60)

Line 55: "infectious to humans" --> better "infecting humans"

Reply: We edited accordingly (line 63)

Line 95: Feeding --> feeding

Reply: We corrected accordingly (line 103)

Line 98: -20 ºC --> -20ºC

Reply: We corrected accordingly (line 106)

Line 102: "of the suspension of stool...", better "of the stool suspension..."

Reply: We changed accordingly. (line110)

Line 105-106: 60 ul together, not in different lines. -70 ºC --> -70ºC

Reply: We edited accordingly (line 112-113).

Line 107: "The presence of NoVs..." --> "The presence of NoV..."

Reply: We corrected accordingly (line 118)

Line 110: "and a 2 ul virus RNA extract.." --> "and 2 ul viral RNA extracted.."

Reply: We changed accordingly

Line 114: "of 650bp..." --> of 650 bp..."

Reply: We changed accordingly (line 125)

Line 138: "infections by using..." --> "infections by..."

Reply: We removed “using” accordingly (line 142)

Line 139: "RT-PCR by using..." --> "RT-PCR using..."

Reply: We removed “by” accordingly (line 143)

Line 141: "was done _ by..." -->  "was done by..." 

Reply: We corrected accordingly (line 144)

Line 166: "from participants aged between 0-59 months..."

Reply: We added “between” accordingly (line 173)

Line 171: "...lived an urban..." --> "...lived in an urban..."

Reply: We corrected accordingly (line 178)

Line 282: "...sequences of P2-region..." --> "...sequences of P2 region..."

Reply: We corrected accordingly (line 289)

Line 311: "...a 593bp..." --> "...a 593 bp..." 

Reply: We corrected accordingly (line 318)

Line 322: "...was constructed on the base of 459 bp nucleotide sequence of ..."

Reply: We changed according (line 329-330)

Line 329: "...of which 23.2% (41/177) of symptomatic..." --> "...of which 23.2% (41/177) were symptomatic..."

Reply: We changed accordingly (lines 336-337)

Line 390: "All detected the AiV..." --> "All detected AiV..." 

Reply: We corrected accordingly (line 403)

While hoping to have addressed your concerns adequately and looking forward to a favourable response, we remain ready to provide any further clarifications.

Yours Sincerely,

Dr. Sandra Niendorf

Reviewer 2 Report

Overall, this is a well-organized manuscript. However, in the discussion section, the importance of the results should be dealt with in more depth.

Author Response

Dear Reviewer,

We thank you very much for your appreciation of our study.

Comments and Suggestions for Authors

Overall, this is a well-organized manuscript. However, in the discussion section, the importance of the results should be dealt with in more depth.

Reply: We are very thankful for this helpful comment. We added few sentences in the discussion section following the above suggestion (lines 375-380 and line 409-416). 

While hoping to have addressed the reviewer concerns adequately and looking forward to a favourable response, we remain ready to provide any further clarifications.

Yours Sincerely,

Dr. Sandra Niendorf

Round 2

Reviewer 1 Report

Thanks for the changes and improvements.

I still suggest you that maybe a combination of table + figure showing all the data collected, would be easier to see. But I understand what you mean, it is only a suggestion.

Supplementary Table 1.- In the 5th row, last column, should say AstV.

Supplementary Table 2.- First row, last column. Change "Used primers" to Primers.

Author Response

Response to Reviewer 1 Comments

Dear reviewer,

thanks again for your comments to our manuscript.

I still suggest you that maybe a combination of table + figure showing all the data collected, would be easier to see. But I understand what you mean, it is only a suggestion.

Responds: Thanks for your understanding for our decision to leave the table as it is.

Supplementary Table 1.- In the 5th row, last column, should say AstV.

Responds: We have corrected the table as suggested. 

Supplementary Table 2.- First row, last column. Change "Used primers" to Primers.

Responds: Thanks for this helpful comment, we have changed it accordingly.